# Toward Cancer Diagnostics of the Tumor Suppressor p53 by Surface Enhanced Raman Spectroscopy

**DOI:** 10.3390/s20247153

**Published:** 2020-12-14

**Authors:** Anna Rita Bizzarri, Salvatore Cannistraro

**Affiliations:** Biophysics and Nanoscience Centre, Dipartimento DEB, Università degli Studi della Tuscia, 01100 Viterbo, Italy; cannistr@unitus.it

**Keywords:** surface enhanced Raman spectroscopy (SERS), p53, p53 mutants, ultrasensitive detection, cancer biomarkers

## Abstract

The tumor suppressor p53 protein plays a crucial role in many biological processes. The presence of abnormal concentrations of wild-type p53, or some of its mutants, can be indicative of a pathological cancer state. p53 represents therefore a valuable biomarker for tumor screening approaches and development of suitable biosensors for its detection deserves a high interest in early diagnostics. Here, we revisit our experimental approaches, combining Surface Enhanced Raman Spectroscopy (SERS) and nanotechnological materials, for ultrasensitive detection of wild-type and mutated p53, in the perspective to develop biosensors to be used in clinical diagnostics. The Raman marker is provided by a small molecule (4-ATP) acting as a bridge between gold nanoparticles (NPs) and a protein biomolecule. The Azurin copper protein and specific antibodies of p53 were used as a capture element for p53 (wild-type and its mutants). The developed approaches allowed us to reach a detection level of p53 down to 10^−17^ M in both buffer and serum. The implementation of the method in a biosensor device, together with some possible developments are discussed.

## 1. Introduction

The p53 protein, commonly referred as tumor suppressor, is a transcription factor which plays a central role in maintaining the genome integrity of the cell. It is coded by the TP53 gene which belongs to a highly conserved family of genes [1,2]. p53 is controlled by a variety of post-translational modifications and of interactions with several target signaling proteins [3]. In normal conditions, the level of p53 is kept low by a regulatory feedback loop mainly involving mouse double minute 2 homolog (MDM2) and MDMX [4,5]. Under cellular stress conditions, such as DNA damage and oncogenic stresses, p53 is activated giving rise to multiple biological processes that lead to repair genome mutations or to cell-cycle arrest, apoptosis and senescence, if the damage is irreparable [3]. p53 was found to be inactivated in several tumors, through either mutations in the TP53 gene or a deregulation of its associated path [6,7,8,9]. On the other hand, p53 is overexpressed in many tumors, as due, e.g., to specific oncogenic stresses or MDM2 gene alterations. These changes yield a deregulation of p53 associated pathways, with drastic effects on cell health [10].

The presence of mutated p53 proteins in human blood, can be put into relationship to a pathological state [11,12,13]. On the other hand, abnormal concentrations of wild-type p53 (p53_wt_) were found in some cancers [14]. Accordingly, p53 represents a valuable cancer biomarker and its ultrasensitive detection can be an extremely useful tool for non-invasive screening approach in the field of tumor early diagnostics. Additionally, p53 detection can help in evaluating prognosis also in the perspective of next-generation individualized precision medicines. High efforts are therefore devoted to develop biosensors able to selectively reveal p53 at very low concentrations [15]. With such an aim, different techniques, often combined with nanotechnological strategies, were applied. They range from traditional methods, such as enzyme-linked immunosorbent assay (ELISA), electrophoretic immunoassay and mass spectroscopy, to more innovative and advanced techniques, such as surface plasmon resonance (SPR), surface enhanced Raman spectroscopy (SERS), field-effect transistor (FET), chemi-luminescence, calorimetry, quartz crystal microbalance, etc. [16,17,18,19,20,21].

Detection of p53 presents, however, some more difficulties in comparison to those of other proteins because of its structural disorder [22]. Indeed, p53 belongs to intrinsic disorder protein (IDP) family which are characterized by the presence of some disordered regions giving rise to a multiplicity of configurations, often co-existing in solution and resulting in a structural heterogeneity [23,24]. On one hand, the partial lack of well-defined structure confers to IDP a high adaptability in terms of binding capability to different partners; but, on the other, it may limit the strength of the formed complexes [25]. For an effective biosensing capability, a special care should be then devoted to the choice of the biomolecular capture element, able to form a stable complex with p53, through a specific biorecognition process.

In our group, we investigated the interaction of p53, and also of other molecules of p53 family (p63 and p73) with various ligands, such as Azurin (Az), Az-derived peptides, MDM2, E3 ubiquiting-protein ligase COP1 [26,27,28]. These studies were performed by different techniques, operating in bulk or at single molecule level, such as atomic force microscopy and spectroscopy (AFM and AFS), SPR, fluorescence, Foester resonance energy transfer (FRET) and Raman [29,30,31,32]. A particular attention was paid to the interaction of p53 with the redox blue copper-containing protein Az from the bacterium *Pseudomonas aeruginosa* which has been shown to exert an anticancer action, upon its binding to p53 [33,34].

The study of the p53 binding network was combined with the previous and consolidated experience on ultrasensitive detection of different biomarkers by SERS [35,36,37], to implement highly specific and sensitive detection of p53 [38,39,40]. SERS is a Raman-based technique exploiting the huge enhancement of the Raman cross section when molecules are placed in close proximity of a nanostructured metal surface, typically aggregated gold and silver nanoparticles (NPs). Hence, SERS conjugates the richness of chemical and structural specificity of Raman spectroscopy with a very high sensitivity, and then, it is an extremely suitable technique to detect molecules at very low concentration [41,42,43]. However, to effectively reach notable results, SERS-based approaches should be developed in a close connection with suitably functionalized nanostructures, specifically tailored for the biomarker to be detected.

Here, we reviewed and revisit our SERS-based approaches for ultrasensitive detection of wild-type and some mutants of p53, in the perspective to develop biosensor devices to be used in clinical applications [38,39,40]. Our approaches exploit nanotechnological strategies, with the help of surface chemistry, to prepare suitable gold NPs and substrates; both of them being functionalized with one of the two partners. The Raman marker is provided by a small molecule (4-ATP) acting as a bridge between gold NPs and a protein molecule; the corresponding Raman signal being highly enhanced when 4-ATP molecules are conjugated to gold NPs. Functionalized NPs were fluxed on the substrate to promote a biorecognition process between the molecular partners and then the capture from the substrate. The final system was scanned under the microscope objective for detection. Different molecular architectures, together with the use of different capture elements for targeting p53, were investigated. In particular, we exploited the capability of Az, or specific monoclonal antibodies, to bind both p53_wt_ and some p53 mutants. For each system, the molecular strategy was carefully refined in order to maximize the detection capability. The results were evaluated in terms of sensitivity, selectivity, accuracy and reproducibility; the potentialities of the best detection method to be implemented in clinical biosensors were also analyzed. Finally, our results were discussed in connection with other available SERS-based approaches for ultrasensitive detection of p53 and possible developments were briefly presented.

## 2. Biosensor: Detection Technique, Molecular Components and Procedures

### 2.1. Biosensors: Basic Principles

A biosensor is a device which combines a biological component (capture element) able to recognize a specific analyte (target) and then producing, through a physico-chemical component (transducer), a measurable signal for detection. The main features required for a biosensor are: (i) the accuracy; (ii) the reproducibility; (iii) the stability with respect to the measurement conditions; (iv) the Limit of Detection (LOD), which is the lowest concentration that can be detected; (v) the sensitivity, given by the variation of the biosensor response with respect to the variation of target concentration; (vi) the operating range, providing the concentration range of detection; and finally (vii) the selectivity which should be evaluated with a particular attention with respect to possible competitors of the target. Additionally, a linear response with concentration is commonly required. In the following, our SERS based detection methods will be presented and discussed in connection with the main biosensor features also in the perspective to be implemented in a device for real applications.

### 2.2. SERS Principles, Equipments and Measurement Conditions

As already mentioned, SERS is based on the huge enhancement, up to several orders of magnitude, of the extremely weak Raman cross-section, when molecules are close to a nanostructured metal surface (mainly noble metals) [44]. Such an effect is generally attributed to two main mechanisms: (i) an electromagnetic enhancement (EM) mechanism, associated with the occurrence of large local field caused by surface plasmon resonance; and (ii) a charge transfer chemical mechanism (CT) from the metal to the adsorbed molecules [45]. The real underlying processes are characterized by a rather complex interplay among them, and the signals may fluctuate in time [46,47]. Accordingly, some effort should be devoted to establish stable measurement conditions with a maximized enhancement. Different regions (3 × 3 mm^2^) of each substrate sample were manually scanned by the microscope objective in order to localize sites characterized by extremely high scattering; these sites, called “hot spots”, being usually related to highly Raman enhancement. For other experimental details see refs. [40,48].

### 2.3. Target Molecules: Wild-Type and Mutated p53

p53 is a tetrameric protein in which each monomer is made of 393 amino acid residues, arranged into four functional regions: the N-terminal transactivation (NT) domain, the DNA binding domain (DBD), the tetramerization domain and the C-terminal regulatory (CT) domain. As due to the presence of some disordered regions, the structure of the full-length p53 is still unresolved [48,49]. In the following, we considered wild-type p53 (p53_wt_), and three single point mutants: p53_R249S_, p53_C135V_ and p53_R175H_, with a mutation inside the DBD portion. All the mutants are valuable biomarkers due to their high relevance in cancer.

Figure 1a shows a graphical representation of the DBD portion of p53_wt_ from 1 TUP entry of the protein data bank (PDB) [50]; the positions of amino acids susceptible to be mutated in our investigation are marked (see legend of Figure 1). The single mutation in p53_R249S_ (azur sticks in Figure 1a), yields a structural distortion of in the L3 loop (residues 243–249) of DBD affecting then its binding capability to DNA [51]; this mutation being very commonly observed in cancer [52]. In particular, it was found to be strictly related to mutagenesis as induced aflatoxin [53].

The p53_C135V_ mutant (green sticks in Figure 1a) is a temperature-sensitive mutant, involved in neoplastic cell formation, tumor invasiveness, and genotoxic stresses [56]. Below 37 °C, p53_C135V_ behaves as the wild-type p53, while at higher temperature its conformation switches to a mutant phenotype. Finally, the p53_R175H_ mutant (blue sticks in Figure 1a) is found in several tumors, such as lung, colon and rectal and with a particularly high incidence in breast tumor [57,58,59].

### 2.4. Capture Elements for p53

The capture element of a biosensor is tasked to selectively and stably bind the target (here wild-type or mutated p53) through a biorecognition process. As already mentioned, such a task could be rather difficult for p53, as due to the presence of some unstructured regions which may result into some instability [22]. As capture molecules for p53, we selected two different kinds of molecules: Az and specific antibodies. The blue copper-containing Az protein (MW 14kDa), whose structure from X-ray investigation is represented in Figure 1b [54], is characterized by a high structural stability and well-defined mechanical properties [60,61]. Notably, Az exhibits peculiar redox and optical properties which make it suitable also for bio-nano electronics applications [62,63,64]. As already mentioned, Az exerts an anticancer action, by enhancing apoptosis in tumor cells upon its binding to p53 [65]. For the Az-p53 complex, we found a dissociation rate value of k_off_ of 0.09 s^−1^ and a dissociation constant K_D_~6x10^−6^ M [65]. Furthermore, a tentative binding region between Az and DBD was also proposed by applying computational docking with the help of mutagenic data [66,67]; the amino acids expected to be involved in the DBD/Az complex being marked in red in Figure 1b. Az was also demonstrated to selectively bind both the p53_R249S_ and p53*_C135V_* mutants [39]. These features make Az an extremely valuable capture element for p53. As alternative capture biomolecules, we used monoclonal antibodies, specifically selected to target the p53_wt_ or the mutants of interest. In particular, we used the mouse monoclonal anti-p53_wt_ PAb 1620 antibody (anti-p53_wt_, MW 150 kDa) and the rabbit polyclonal anti-p53_R175H_ antibody (anti-p53_R175H_, MW 150 kDa) [40]. Finally, suitably designed aptamers can be used.

### 2.5. Description and Characterization of the 4-ATP-NP Probe

The structure of the 4-aminothiophenol (4-ATP) molecule, used as Raman marker in our detection method, is shown in Figure 2.

The Raman spectrum of 4-ATP, as obtained by an excitation of 633 nm, is shown in Figure 3a. In the 1000–1500 cm^−1^ region, the spectrum exhibits three main bands, with other small peaks being hardly detectable. At 2555 cm^−1^, a band, attributed to the stretching vibration of –SH group of 4-ATP, is also present (see the inset in Figure 3a) [68]. The assignment of the main bands to the 4-ATP vibrational modes is reported in refs. [39,68]. The Raman spectrum of 4-ATP as obtained by using an excitation wavelength of 514 nm is almost the same (not shown) [40].

The Raman-SERS spectra of 4-ATP upon their conjugation to 50 nm gold NPs (4-ATP-NP), excited at 633 nm and 514 nm, are shown in Figure 3b,c, respectively. In both the spectra, the band centered at 2550 cm^−1^ disappears (see the insets), witnessing the formation of a covalent bond between the 4-ATP thiol group and gold (S-Au bond). The Raman spectra in Figure 3b,c substantially show the same peaks, although with a slightly different relative intensity. Some of these peaks are also observed in the spectrum of 4-ATP alone, even if with a small red-shift, and a different relative intensity; for the assignment of the main peaks of 4-ATP when conjugated to a gold NP see ref. [68].

The enhancement of the 4-ATP signals conjugated to gold NP, with respect to 4-ATP alone, was determined from the signal to noise (S/N) ratio of the intensity of a select peak (reference signal) to that of the noise signal. As reference signal, we selected the peak of the C-S stretching mode, characterized by a strong Raman signal and detected at 1089 cm^−1^ and 1078 cm^−1^ for 4-ATP and 4-ATP-NP, respectively. The noise signal was determined from the intensity averaged in (1800 ± 10) cm^−1^ spectral region, where no Raman signals are present [39]. The average of the S/N ratio evaluated from ten spectra independently prepared samples, resulted into a SERS enhancement of about seven orders of magnitude with an excitation at 633 nm, and of six orders with an excitation at 514 nm.

The conjugation of 4-ATP with NPs was more closely monitored by following the reaction between 4-ATP and NP with optical spectroscopy.

The absorption of bare gold NP, shown in Figure 4a (black line), reveals a band centered at about 533 nm arising from the surface plasmon excitation. Upon binding of 4-ATP to gold, the band undergoes a slight broadening together with an upward shift of the peak up to 537 nm (red line in Figure 4a); with this witnessing the formation of a bond between 4-ATP and NP (4-ATP-NP) [68,69]. The shift was found to progressively increase as far as more 4-ATP molecules are bound. To fulfill the requirements of a high covering of NPs with 4-ATP, optimized conditions were searched by varying the concentration of 4-ATP, the temperature and the reaction time (see also below).

The 4-ATP-NP system was charged with one of the biomolecules of interest (Azurin, p53 antibody or p53) through a diazo coupling reaction, as sketched in Figure 2 [35,70]. Briefly, the free end of 4-ATP-NP (the -NH_2_ group) was first activated to give rise to the diazonium compound (−N^+^≡N), able to react with the exposed histidine or tyrosine residues of a protein molecule, forming a diazo bond (N=N) [68]. Such a reaction leads to the 4-ATP-NP tightly linked a protein, constituting the probe, named 4-ATP-NP-pro. To avoid the presence of unbound reactive sites on the NP surface, the 4-ATP-NP-pro was further treated with ETA (for more details see ref. [40]).

Figure 4b representatively shows the SERS spectrum of the 4-ATP-NP probe, as charged with anti-p53. A comparison with the SERS spectrum of 4-ATP-NP alone (see Figure 3) indicates the appearance of an additional strong Raman band centered at about 1328 cm^−1^ (highlighted in Figure 4b). Such a band can be assigned to the stretching vibration modes of the diazo bond and it indicates the formation of a covalent bond between 4-ATP and the biomolecule. Notably, such a band is the same irrespectively of the specific protein conjugated to the 4-ATP-NP; in our procedures the protein can be Az, p53 antibody or p53. The presence of a well distinguishable vibrational fingerprint, combined with a very high enhancement by the plasmonic effect, make the 1328 cm^−1^ band an appropriate and very efficient Raman marker (called also Raman fingerprint) to be monitored in SERS-based detection methods [68].

The number of biomolecules effectively conjugated to 4-ATP-NP, useful to find out optimized conditions for the probe, can be estimated by fluorescence experiments [71]. The initial solution containing the protein (here anti-p53_wt_), used in the probe assembly and the supernatants, after successive incubation steps, were investigated by static fluorescence emission. The signal obtained by an excitation at 280 nm and collected at 340 nm, was followed. Fluorescence emission intensity of the supernatants progressively decreases as consequence of the removal of unbound molecules. The amount of the molecules conjugated to 4-ATP-NP (C_conjugated_), can be then evaluated from the following expression:(1)$$C_{conjugated}=C_{0}-C_{unbound}=C_{0}\;\frac{F_{0}-F}{F_{0}}$$
where *C*_0_ and *F*_0_ are the concentration and fluorescence intensity of the initial solution, respectively; while *C_unbound_* and *F* are the concentration and the fluorescence intensity of the supernatant, respectively. We found that about 2.5 × 10^3^ molecules of anti-p53_wt_, corresponding to 1.9 × 10^−7^ M, were conjugated to a single NP. By applying the same procedure, we found that about 250 and 980 molecules of p53 and Az, respectively, were conjugated to a single NP.

### 2.6. Description and Characterization of the Capture Substrate

The functionalization procedure of the glass substrates for capturing the 4-ATP-NP probe was done by following the steps sketched in Figure 5 and reported in ref. [40].

Briefly, after a cleaning procedure, the glass substrates were irradiated by ultraviolet (UV) light to increase the number of silanol groups on the glass surface. They were then treated with 3-aminopropyltriethoxy-silane (APTES) dissolved in chloroform and reacted with 1% glutaraldehyde solution (GLUTA). The substrates were then incubated with a drop (40 µL) of solution containing the biomolecules at the required concentration; with this promoting the formation of a self-assembled monolayer through the solvent-exposed amino groups. In the approach using antibodies as capture elements, the substrates were also treated with the glutathione S-transferase (GST) protein to block eventually unreacted sites. Finally, the functionalized substrates were incubated with a drop (10 µL) of the probe at 4 °C, to favor the capture of the molecule conjugated to the 4-ATP probe through a biorecognition process. The incubation times were in the 2–10 h range in order to maximize the number of hot spots showing the features of the 4-ATP-NP probe. The substrates were finally rinsed to remove uncaptured probes. Blank reference substrates were prepared by following the same procedure, but skipping the incubation step with the biomolecules. For more details see ref. [40].

## 3. Detection of p53 by SERS

### 3.1. Overview

Different methods for ultrasensitive detection by SERS of wild-type and mutated p53, in buffer and in serum, were described and analyzed in the perspective of an implementation into a biosensor. All the methods exploit the active molecule (4-ATP) conjugated, on one end to gold NPs, and, on the other, to a protein molecule (target or capture element); the resulting system, named 4-ATP-NP-pro, constituting the SERS probe. Indeed, the Raman signal, at 1328 cm^−1^, emerging from the conjugation of 4-ATP to the protein, provides the vibrational fingerprint to be followed for detection. The SERS probes are fluxed onto a substrate on which the biomolecular partners were previously immobilized. The final sample was then scanned by the microscope objective to reveal the Raman fingerprint arising from probes captured by the substrate through a biorecognition process.

### 3.2. Detection of p53 in Buffer Using Az

The procedure, called 4-ATP-NP-p53/Az, to detect p53 in buffer involves a probe functionalized with the target (p53) and a substrate charged with the capture molecule Az. Figure 6a shows representative SERS spectra of the substrate before and after incubation with the probe; the substrate having been charged at progressively lower concentrations of p53_wt_ (from 5 × 10^−11^ M to 5 × 10^−13^ M). The absence of any detectable Raman signal in the analysed region in the substrate before incubation with the probe indicates that the substrate itself does not interfere with our Raman analysis. At variance, different peaks appear in the spectra corresponding to samples containing p53_wt_. In particular, it is evident the presence of the Raman marker at 1328 cm^−1^ (marked in yellow in Figure 6); this reflecting the capture of the probes from the substrate through a biorecognition process between Az and p53_wt_. At visual inspection, the intensity of the Raman fingerprint decreases by lowering the concentrations of p53, in agreement with a reduction of hot spots. Such an aspect was, however, quantitatively analysed by evaluating the S/N ratio which exhibited a progressive decrease as far as lower p53 concentrations were taken into consideration. Additionally, the S/N was found to become less than 3 (at which the signal is assumed to be undetectable over the noise), for concentrations below 5 × 10^−13^ M (not shown); such a value being then assumed as the LOD for p53_wt_ detection in buffer. As a control, we performed an experiment in which 4-ATP-NPs (at a concentration of 1 × 10^−12^ M) not charged with p53, were deposed on the Az-coated substrate. In this case no signal over the noise was detected, with this indicating that no capture events occur without the target.

The same method was applied for detection of p53_R249S_ and p53_C135V_ mutants. Figure 6b shows the SERS spectra of the substrate, before and after deposition of the 4-ATP-NPs charged with p53_R249S_ at concentrations from 5 × 10^−11^ M to 5 × 10^−13^ M. The Raman fingerprint is clearly evident down to 5 × 10^−13^ M, as confirmed by a S/N noise analysis. At lower concentrations, it is no more detectable over the noise. Similar results were obtained for the p53*_C135V_* mutant (not shown). For both the p53_R249S_ and p53*_C135V_* mutants in buffer, a LOD of 5 × 10^−13^ M was found. Accordingly, it can be inferred that Az is a suitable capture element for p53_wt_ as well as for the two p53 mutants. Such a finding is consistent with the fact that the analyzed mutations do not interfere with the involved Az-p53 binding site. Indeed, the capability of Az to bind p53, irrespectively of the presence of these two mutations, hinders the possibility of discrimination among p53_wt_, p53_R249S_ and p53*_C135V_*. When such a task is required, other approaches should be used.

Finally, we explored a reversed strategy, in which the substrate was functionalized with p53, while the 4-ATP-NP was charged with Az molecules. Although in principle such a strategy should be better because a higher number of Az molecules can be bound to a single 4-ATP-NP (980 for Az against 250 for p53), almost the same results in terms of LOD were obtained. This supports that the reached detection level of the method depends on the contribution from different factors whose optimization requires a careful global investigation.

### 3.3. Detection of p53 in Serum Using Az

The capability of Az to detect p53_wt_ or p53_R249S_ in serum was then investigated. In this case, the large presence of the other molecules (lipids phosholipids, and other proteins, such as albumin and globulins, etc.), might interfere with the Az-p53 interaction, limiting the p53 detection. We selected the configuration in which Az was conjugated with 4-ATP-NP and p53 charged on the substrate, since it resulted more efficient with respect to the reversed one. Accordingly, substrates were functionalized with pure human healthy serum and serum spiked with p53_wt_ or p53_R249S_ at different concentrations.

From Figure 7, it is evident that no Raman fingerprint over the noise was detected for samples obtained incubating the probe with a substrate functionalized only with pure human serum; with this indicating that Az does not significantly interact with the molecules deposed on the substrate. At variance, the peak at the Raman fingerprint at 1328 cm^−1^ appears in the spectra obtained by flowing the probe on substrates functionalized with serum spiked with p53_wt_ or p53_R249S_, at 5 × 10^−10^ M, 5 × 10^−11^ M and 5 × 10^−12^ M; spectra related to p53_R249S_ being shown in Figure 7. Other Raman bands corresponding to 4-ATP-NP can be also observed in these spectra, confirming therefore its presence within the substrate. At lower concentrations, no signal over the noise is detected. Accordingly, a LOD of 5 × 10^−12^ M can be derived for detection of p53_R249S_ in serum. Similar results were obtained for the p53_wt_ (not shown). Therefore, this method is able to specifically identify both p53_wt_ and the p53_R249S_ mutant in a serum-like environments, where many possible interference components are present, with a LOD of 5 × 10^−12^ M.

### 3.4. Detection of p53 in Buffer Using Abp53

The detection procedure 4-ATP-NP-Abp53/p53_wt_, exploits a specific monoclonal antibody (Abp53) as a capture element for p53. In this case, we used a configuration in which the 4-ATP-NP was functionalized with Abp53, while the substrate was charged with p53 at different concentrations. Figure 8a shows the SERS spectrum from the capture substrate functionalized with p53_wt_ (at 10^−12^ M) before incubation with the probe. No Raman signal over the noise can be observed. Figure 8a also shows the SERS spectra from substrates functionalized with p53_wt_ at different concentrations (the 10^−12^, 10^−15^ and 10^−17^ M) after incubation with the 4-ATP-NP-Abp53_wt_ probe. The Raman fingerprint at 1328 cm^−1^ (marked in yellow), can be clearly detected in all the spectra witnessing the presence of the 4-ATP-NP-Abp53_wt_ probe as due to a biorecognition process between Abp53 and p53_wt_. More specifically, the Raman fingerprint is characterized by a S/N signal higher than 3, for a target concentration down to 10^−17^ M, while it becomes practically undetectable over the noise, at lower concentrations. Accordingly, a LOD of 10^−17^M was assumed for detection of p53_wt_ by this method.

Similar results were obtained by incubating the 4-ATP-NP-Abp53_R175H_ probe with substrates charged with p53_R175H_, in the 10^−12^–10^−17^ M concentration range and shown in Figure 8b. Again, the characteristic Raman fingerprint at 1328 cm^−1^ (marked in yellow in Figure 8b) can be clearly observed down to 10^−17^ M; such a concentration indicating the LOD of this method. As a further control, the 4-ATP-NP-Abp53_wt_ probe (or the 4-ATP-NP-Abp53_R175H_ probe) was incubated on the blank substrate. No Raman signal over the noise were found in both cases (not shown). This means that the probes do not undergo a significant interacting with the blank substrate.

The possibility of a cross-reactions between p53_wt_ and p53_R175H_ was evaluated by analyzing the following samples: (i) the 4-ATP-NP-Abp53_wt_ probe combined with the substrate functionalized with p53_R175H_; and (ii) the 4-ATP-NP-Abp53_R175H_ probe combined with the substrate functionalized. In both the case the substrate was charged the target at a concentration of 10^−15^ M. The characteristic Raman fingerprint at 1328 cm^−1^ was not detected over the noise level from the Raman spectra of both the samples (not shown). These results therefore demonstrate that this method is able to well discriminate between the p53_wt_ and p53_R175H_. Remarkably, the use of specific antibodies of p53_wt_ allowed us to reach a lower detection level of p53_wt_ in comparison to that using Az. Such a sensitivity improvement in the for p53_wt_ detection can be mainly attributed to a more specific recognition by the antibody with respect to Az; however, a contribution from other aspects, such as the optimized immobilization strategy, cannot be ruled out. In particular, the use of GST to block unreacted sites could help in maximizing biorecognition events.

### 3.5. Detection of p53 in Serum Using Abp53

To apply the developed method to the detection of p53 in serum, we preliminarily analysed the substrate functionalized with bare healthy human serum. The Raman spectrum, shown in Figure 9, reveals the presence of two broad bands at about 1050 and 1175 cm^−1^. These bands were not observed in the substrate functionalized only with p53 (see Figure 8), they are likely attributable to the various biomolecules present in serum (lipids phosholipids, and other proteins, such as albumin and globulins, etc.), and deposed on the substrate. However, the absence of Raman signals in the 1328 cm^−1^ region allowed us to use our probe for detection. Notably, these bands were not detected by the Az-based method (see Figure 7) likely due to the slightly different functionalization procedures (see Section 2.6).

Then, we analysed the samples obtained by incubating the 4-ATP-NP-Abp53_wt_ probe (or 4-ATP-NP-Abp53_R175H_) on the substrates previously functionalized with human serum spiked with p53_wt_ (or p53_R175H_) at different concentrations. We found that the Raman fingerprint at 1328 cm^−1^ is clearly detectable for both p53_wt_ and p53_R175H_ samples, down to 10^−15^ M; representative SERS spectra for p53_wt_ being shown in Figure 9. In all the cases the S/N ratio was found to be higher than 3, while at lower concentrations, the Raman fingerprint is no more detectable over the noise. Consequently, the 10^−15^ M concentration constitutes the LOD for the p53_wt_ and p53_R175H_ detection in human serum. The cross-reaction between our probes (Abp53_wt_ or Abp53_R175H_) and the healthy human serum constituents, was checked. The spectrum obtained by incubating the 4-ATP-NP-Abp53_wt_, probe on substrate functionalized with healthy human serum, does not reveal any significant signal at 1328 cm^−1^ over the noise. This means that the probe was not captured by the substrate. A similar spectrum was obtained by dropping the 4-ATP-NP-Abp53_R175H_ probe on the bare healthy human serum-functionalized substrate.

Additionally, the possibility of some interference between p53_wt_ and p53_R175H_, was investigated by taking into consideration the samples in which the 4-ATP-NP-Abp53_wt_ probe was used in connection with serum spiked with both p53_wt_ and p53_R175H_; with both the targets were at a concentration of 10^−15^ M. We found a spectrum almost the same as that obtained from the serum sample in which only p53_wt_ was present (Figure 9). The same results were obtained when the 4-ATP-NP-Abp53_R175H_ probe was used with serum spiked with both p53_wt_ and p53_R175H_. These results indicate that our approach is able to well discriminate between the p53_wt_ and p53_R175H_. In summary, the presented method is able to selectively detect both p53_wt_ and p53_R175H_ proteins down to 10^−15^ M in human serum.

### 3.6. Comparison with Other SERS-Based Detection Methods for p53

As already mentioned, several different approaches were applied for detection of p53 (wild-type and mutated forms). An overview of these methods, together with the related advantages and limitations, were recently presented (see ref. [15]). On the other hand, although a large variety of SERS-based approaches were developed for detection of several different biomolecules, a very few approaches were developed for p53. Here, we briefly reviewed some SERS-based methods to detect p53 available in literature.

The main features and the detection results of these methods are summarized in Table 1; for comparison data from ELISA, which represents a benchmark, being also reported. Practically, almost all the methods use antibodies as capture elements, however, they are conjugated with rather different molecular strategies; all of them having been able to reach a rather low detection level. As emerging from Table 1, among other methods, our approaches combine the capability of a very low LOD value, with a wide concentration range. In particular, the antibody-based method can reach a very competitive detection level even with respect to the other techniques.

## 4. Analysis of the Performance Parameters in the Perspective to Build a Biosensor for p53

The biosensing approach using antibodies combined with an optimized immobilization strategies, resulted to be the most efficient one to detect p53 in terms of detection level. Indeed, we evaluated, a LOD of 10^−17^ M in buffer and of 10^−15^ M in serum; with these values being comparable with the lowest values available in literature for p53 even by other techniques (see e.g., ref. [15]). Furthermore, the method is also characterized by a good selectivity and accuracy. On such a basis, it could be a suitable candidate to be implemented in a biosensor for detection of wild-type and mutated p53 for clinical uses. In the following, the most relevant performance parameters (LOD, reproducibility, sensitivity, linearity, etc.) of this method were summarized in the perspective of applicative developments.

To test the reproducibility of our detection method, we analysed the peak of the Raman fingerprint by considering five independently prepared samples; for each of them, the integrated area of the Raman band peaked at 1328 cm^−1^, of ten spectra from different regions, was evaluated. The average, and the corresponding standard deviation of this area, for the five samples of p53_wt_ at 10^−15^ M and for p53_R175H_ at 10^−15^ M, in buffer and in serum, are shown in Figure 10a. For each case, the five area values are the same within the errors. This assesses the reproducibility of the detection method for both p53_wt_ and p53_R175H_ in the different environment. Notably, the values for p53 in serum are significantly lower (about 400 a.u.) than those obtained in buffer at the same concentration (about 600 a.u.). Such a decrease could be ascribed to a reduction of the number of specific binding sites available in the substrates functionalized with serum, whose molecules may partially hinder the interaction between p53 and the 4-ATP-NP probe.

To obtain the sensitivity of our approach, a calibration plot was derived by using the integrated area of the Raman marker, peaked at 1328 cm^−1^. The averages and the corresponding standard deviations of this area are plotted as a function of the logarithm of the concentration (from 10^−10^ M and 10^−17^ M), for p53_wt_ (squares) and p53_R175H_ (open circles) (see Figure 10b). We found a very good linear relationship in the whole concentration range for both the cases. In Figure 10b, the resulting fitting curves (continuous lines) by the equation Y = A + BX, where Y is the averaged area intensity and X is the logarithm of the corresponding p53 concentration, are also shown. The goodness of the fit was assessed by the chi-square test. The parameters extracted from the fit are: A = (2520 ± 110) a.u. and B = (133 ± 10) a.u./decade for p53_wt_ and A = (2300 ± 180) a.u. and B = (120 ± 14) a.u/decade for p53_R175H_; with the slope B of the calibration plot providing the sensitivity of the method. Although the slope average appears slightly higher for p53_wt_ with respect to p53_R175h_, the sensitivity is, within the error, almost the same.

The calibration plot was then used to evaluate the capability for our biosensor to quantify p53 samples at unknown concentration. With such an aim, we determined the integrated area of the 1328 cm^−1^ peak from the SERS signal from two samples of p53_R175H_ at the unknown concentrations (named US1 and US2). Upon evaluating the average of the area, the corresponding concentration were extracted from the calibration plot (see stars in Figure 11b). We found C_US1_ = (2.0 ± 0.6)10^−12^ M and C_US2_ = (16.2 ± 0.8)10^−12^ M. The same samples were tested by applying the photometric one-step-enzyme immunoassay p53 ELISA kit for the quantification of unknown samples of p53_R175H_ (100 µl, diluted 1:5 in sample diluent) (see ref. [40]). We found: C_US1_(ELISA) = (1.8 ± 0.4)10^−12^ M and C_US2_(ELISA) = (15 ± 3)10^−12^ M; with both of them resulting statistically consistent with the values obtained by SERS experiments. Such an agreement validates the accuracy of our SERS-based method in the quantification of p53 concentration. In summary, our SERS-based method satisfies a set of requirements which make it suitable for applications in clinical diagnostics.

## 5. Possible Development: Use of Aptamers as Capture Elements for p53

As a possible development of our biosensing approach for detecting p53, we focused our attention to decrease the LOD for the p53_R175H_ mutant, whose presence, even at very small amount, could signal a disease at very early stage [59]. With such an aim, we preliminarily investigated the feasibility of using aptamers, instead of antibodies (or Az), as capture molecules for p53. Aptamers are synthetic molecules, often oligonucleotides, designed to specifically bind a target upon undergoing a biorecognition process [74]. In the following, a preliminarily study finalized to the implementation of aptamers in our SERS-based approach is outlined. For detection of the p53_R175H_ mutant, we can take advantage of the results as obtained by a SELEX procedure which identified a specific aptamer (called here APT175) (see ref. [75]), having a strong affinity for p53_R175H_ and a much less for p53_wt_. Such an aptamer is also able to restore the anticancer activity of mutated p53. Starting from the available sequence of APT175 (ATTAGCGCATTTTAACATAGGGTGC) from ref. [75], the corresponding 3D structure was modelled by following the same procedure described in ref. [76] using the RNAFOLD, [77] and the SIMRNA software [78]. The extracted best model for APT175 is shown in Figure 11a.

For detection, we are intended to use the strategy in which the target (p53_R175H_) is immobilized on the substrate, while the capture element is conjugated with 4-ATP-NP to form the probe. Accordingly, a His tag is to be added to one of the APT175 end. The choice of which APT175 end should be charged with the His tag, was done by preliminarily performing a computational docking between the APT175 and the DBD_R175H_ portion of p53_R175H_ using HDOCK [79]. The structure of the DBD_R175H_ portion was derived from the B chain of DBD in complex with a consensus DNA (1 TUP entry from PDB) [51]), by replacing the Arg175 residue with His175 through the I-TASSER suite [80]. Docking was carried out by also requiring the further conditions that the His175 residue of p53 is to be involved at the binding interface. The first ten complex models with the best docking scores were taken into consideration for further analysis. In these models, the ligand (APT175) was found to cluster into two main groups around the receptor (DBD_R175H_), as representatively shown in Figure 11b). In both the cases, the 3′ end of the aptamer results to be available for hosting a His tag without substantial interfering with the p53 binding. The capability of the approach using APT175-His-tag to detect p53_R175H_ is under investigation and preliminarily results are rather encouraging.

## 6. Conclusions

Detection of p53 at very low concentrations represents an important goal for early diagnostics and prognosis of cancer diseases. SERS, combined with suitably devoted nanotechnological architectures, provides a valuable biosensing approach for ultrasensitive detection of p53. The presented methods were able to detect wild-type and single point mutated p53, down to 10^−17^ M in buffer and to 10^−15^M in human serum. Such a detection capability deserves a high interest for signaling abnormal cancer states or other diseases, with high potentialities in nanomedicine. Among the various approaches that have been reviewed, that one using a specific antibody as capture element of p53, combined with an optimized biomolecular immobilization strategy, was found to exhibit the highest sensitivity. At the same time, it is also characterized by a high reproducibility, accuracy, linearity in a rather wide concentration range, and even a high selectivity between p53_wt_ and the p53_R175H_ mutant. All these features make it extremely suitable for its implementation as a biosensing platform for clinical applications. Among possible improvements, the use of aptamers suitably designed for targeting the p53_R175H_ specific mutants, as alternative capture elements for p53, could be envisaged. From a more general point of view, the method can be also adapted to detect other protein biomarkers upon selecting suitable capture molecules and defining the architecture and functionalization procedures for an optimized detection.

## Figures and Tables

**Figure 1 sensors-20-07153-f001:**
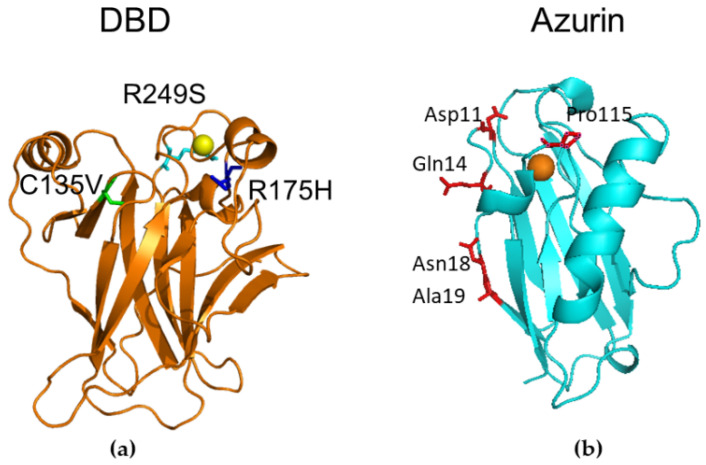
(**a**) Graphical representation of the DBD portion of p53_wt_ from chain B of 1TUP PDB entry [50]. Amino acids undergoing single mutations are marked: p53_R249S_ (azur sticks) p53_C135V_ (green sticks) and p53_R175H_ (blue sticks). Zn ion is represented as a yellow sphere. (**b**) Graphical representation of the blue copper protein Azurin (Az) from chain B of the 1AZU PDB entry [54]. The main amino acids expected to be involved in the interaction with DBD are marked in red. Cu ion is represented as an orange sphere. The figures were created by Pymol [55].

**Figure 2 sensors-20-07153-f002:**
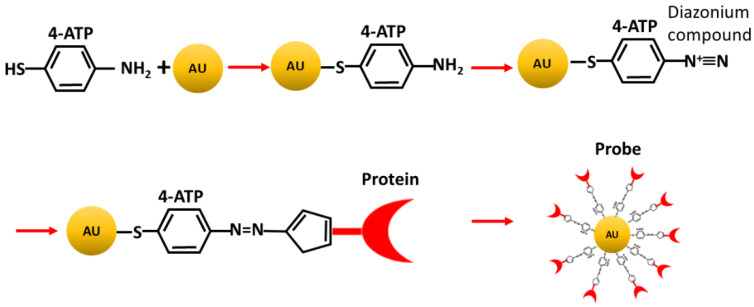
Sketch of the main steps for probe preparation. 4-ATP reacts with gold NP to form the 4-ATP-NP system which is successively activated to yield the diazonium compound able to reacts with a protein target, giving rise to the final probe (4-ATP-NP-pro) (see also the text).

**Figure 3 sensors-20-07153-f003:**
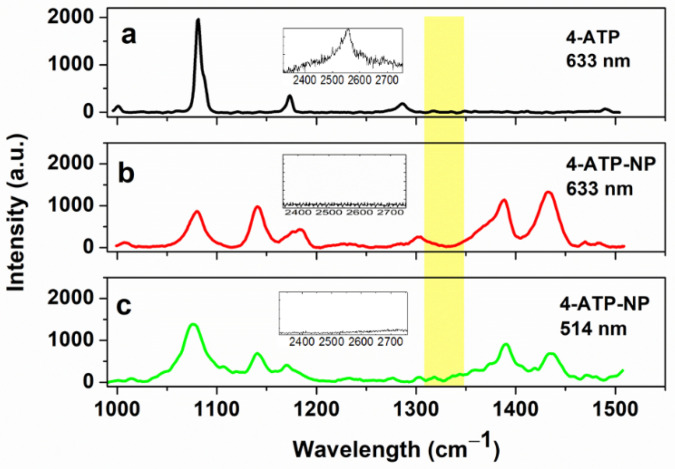
(**a**) Raman spectrum of 4-ATP in water solution obtained with 633 nm laser wavelength. (**b**,**c**) Raman spectra of 4-ATP conjugated to 50 nm gold NPs, obtained with a 633 nm and 514 nm laser wavelength, respectively. The spectral range corresponding to the S–H stretching mode is shown in the insets. The yellow band highlights the region where the Raman marker should appear (not present here).

**Figure 4 sensors-20-07153-f004:**
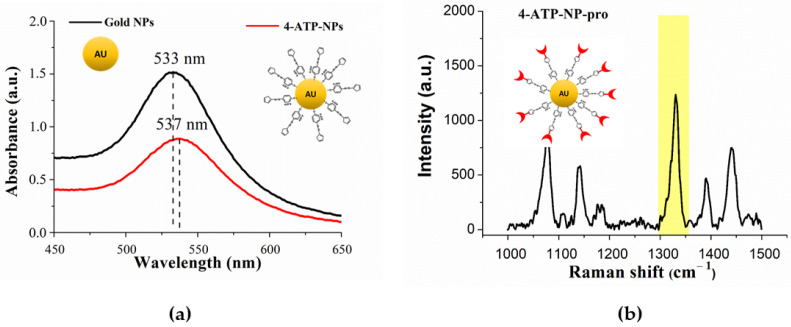
(**a**) Absorption spectrum of NPs before (black line) and after (red line) conjugation with 4-ATP. The maximum absorption peaks are marked by dashed lines. (**b**) Raman spectrum of the probe (4-ATP-NP-pro), sketched in the inset (not in scale), charged with a protein (here antip53). The yellow band highlights the Raman marker peaked at 1328 cm^−1^.

**Figure 5 sensors-20-07153-f005:**
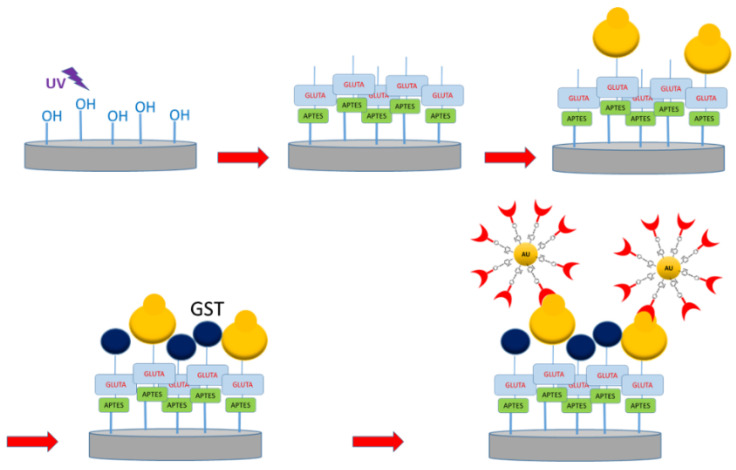
Sketch of the main steps followed for the substrate functionalization. An UV ray treatment to increase the amount of silanol groups is followed by incubation with APTES and GLUTA and then by a functionalization with the biomolecules. Unreacted sites were blocked with GST. Finally, the substrate is incubated with the probe (4-ATP-NP-pro) (see also the text).

**Figure 6 sensors-20-07153-f006:**
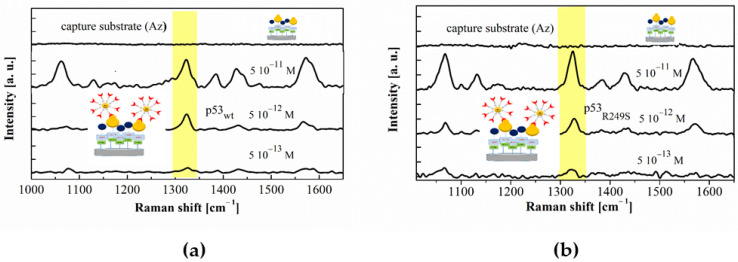
(**a**) Raman spectra of the capture substrate (functionalized with Az and after the incubation with the 4-ATP-NP-p53 probe charged with p53_wt_ at different concentrations. (**b**) Raman spectra of the capture substrate (functionalized with Az and after the incubation with the 4-ATP-NP-p53 probe charged with p53_R249S_ at different concentrations. The yellow band highlights the Raman marker peaked at 1328 cm^−1^.

**Figure 7 sensors-20-07153-f007:**
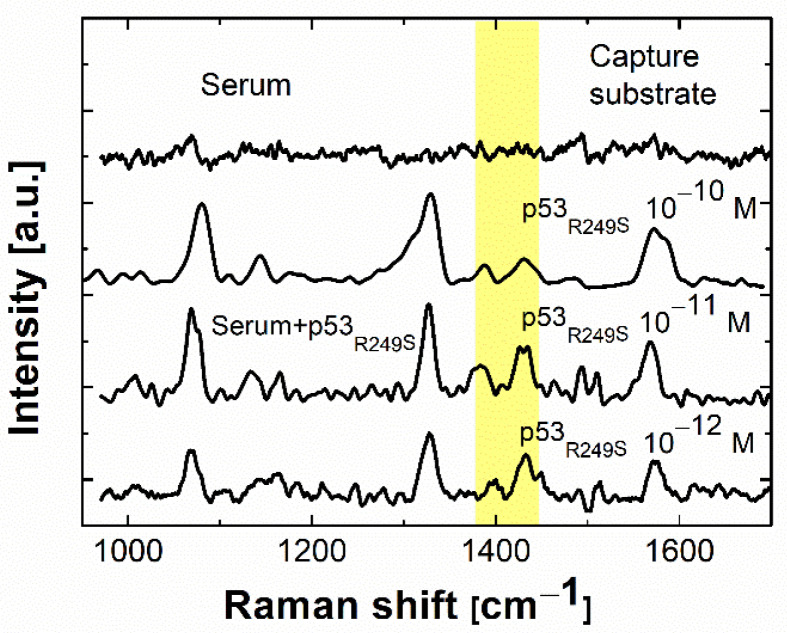
Raman spectra of the capture substrate functionalized with human serum and human serum/p53_R249S_ mixtures at different concentrations of p53_R249S_ and incubated with the 4-ATP-NP-Az probe. The yellow band highlights the Raman marker peaked at 1328 cm^−1^.

**Figure 8 sensors-20-07153-f008:**
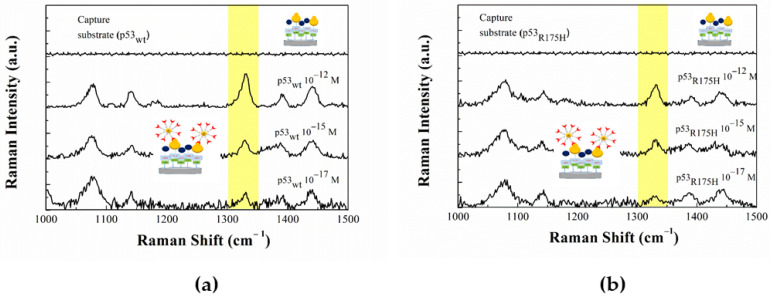
(**a**) Raman spectra of the capture substrate, functionalized with p53_wt_ at different concentrations, before and after incubation with the 4-ATP-NP-pro probe charged with antip53_wt_. (**b**) Raman spectra of the capture substrate functionalized with p53_R175H_, at different concentrations, before and after the incubation with the 4-ATP-NP-pro probe charged with antip53_R175_. The yellow band highlights the Raman marker peaked at 1328 cm^−1^.

**Figure 9 sensors-20-07153-f009:**
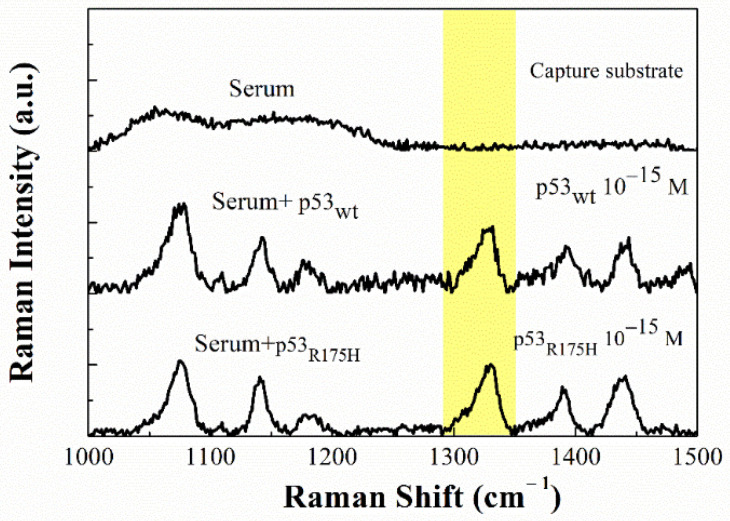
Raman spectra of the capture substrate (functionalized bare healthy human serum) and of functionalized with human serum spiked with p53_wt_ or p53R175H at 10^−15^ M and incubated with the 4-ATP/NP probe charged with anti-p53_wt_. The yellow band highlights the Raman marker peaked at 1328 cm^−1^.

**Figure 10 sensors-20-07153-f010:**
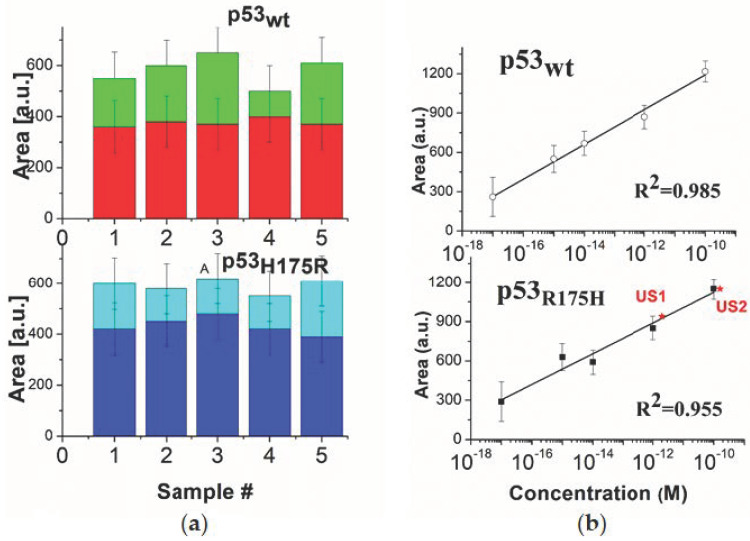
(**a**) Average integrated area of the Raman peak detected at 1328 cm^−1^ for p53_wt_ (top), in buffer (green) and in serum (red), and for p53_R175H_ (bottom) in buffer (azur) and in serum (blue) for five different p53 samples. The average and the corresponding standard deviations (bars) were derived from ten spectra acquired at different regions of the sample. (**b**) Calibration plot obtained from the average integrated area of the Raman peak at 1328 cm^−1^, as a function of the logarithm of the p53_wt_ concentration (full squares) and of p53_R175H_ concentration (open circles). The average and the corresponding standard deviation were obtained from ten spectra acquired at different regions of the sample. The continuous lines represent the fit by the equation Y = A + BX (for the extracted values see the text); the resulting adjusted R^2^ values being reported. The red stars mark the integrated area as obtained for two unknown samples containing p53_R175H_ (see also ref. [40]).

**Figure 11 sensors-20-07153-f011:**
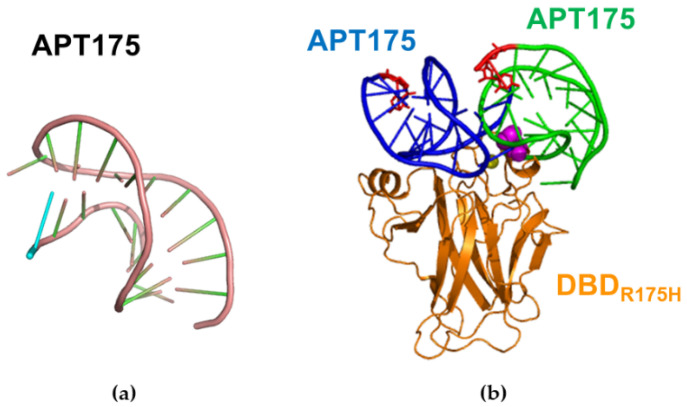
(**a**) Graphical representation of the best model for the APT175 aptamer. (**b**) Graphical representation of the two predicted best arrangements for the APT175 with respect to DBD_R175H_ in the complex. The 3′ end of APT175 is marked in red. The figures were created by Pymol [55].

**Table 1 sensors-20-07153-t001:** SERS-based biosensing methods for detection of p53.

Description	Capture Element	LOD	Concentration Range	References
Au NPs	Azurin	5 × 10^−13^ M	10^−13^–10^−12^ M	[39]
Au NPs	Antibody	1 × 10^−17^ M	10^−17^–10^−12^ M	[40]
Ag substrate	Antibody	5 × 10^−10^ M	10^−10^–10^−6^ M	[72]
Au-Ag Nanorods	Antibody	1 pg/mL	-	[73]
ELISA	Antibody	5 × 10^−12^ M	-	Commercial ELISA

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
