# Peer review of "Toward Cancer Diagnostics of the Tumor Suppressor p53 by Surface Enhanced Raman Spectroscopy"

_sensors, 2020, doi:10.3390/s20247153_

Round 1

Reviewer 1 Report

This paper reviews the authors’ SERS-based methods for ultrasensitive detection of wild-type and some mutated isoforms of p53, in the perspective to develop biosensor devices to be used in clinical applications. The authors’ work on the detection of wild-type and some mutated isoforms of p53 based on SERS is very interesting. However, my main objection is that from the perspective of the whole paper, I’m not sure what the natrue of this paper. Before acceptance for publication I would recommend important changes to be taken inti account in the manuscript. 

There are a number of major issues:

  1. If this paper is reivew, why isthere"Materials and Methods"? Whether is that the structure of the paper should be adjusted and the content be modified accordingly.
  2. The author mentioned that this paperis a review of the authors’ SERS-based methods for ultrasensitive detection of wild-type and some mutated isoforms of p53. Which articles are they? Please quote on line 80 on page 2.
  3. I think this section of “Results and discussion” of this paper are more inclined to be the content of Article rather than Review.

Author Response

  1. If this paper is reivew, why isthere"Materials and Methods"? Whether is that the structure of the paper should be adjusted and the content be modified accordingly.

Although the manuscript was divided into “Materials and Methods”, and “Results and Discussion”, it was conceived as a Review, more specifically as a Feature Article (form which was preliminarily agreed by the Editor) focused on detection of p53 by SERS using the approaches developed by the authors.Accordingly, the article type was changed. In this respect,  I agree that the Section names could be not fully appropriate and then the manuscript was partially reorganized. In particular, the title of the Sections “Materials and Methods” and “Results and Discussion” have been deleted and the related contents were differently grouped.  Additionally, some experimental details have been removed. In particular, Section 2.4  has been shortened (see new Section 2.2), Sections 2.5-2.7  have been grouped (see new Section 2.5) and Section 2.9 has been deleted.

2. The author mentioned that this paperis a review of the authors’ SERS-based methods for ultrasensitive detection of wild-type and some mutated isoforms of p53. Which articles are they? Please quote on line 80 on page 2.

The main references (already cited in the text) have been quoted at Pag.2 Line 79.

3. I think this section of “Results and discussion” of this paper are more inclined to be the content of Article rather than Review.

Besides comments released at Point 1, we would like to remark that the content of the Results and Discussion”  revisits   previous results in which some new aspects are however introduced.

Reviewer 2 Report

In this review article, Bizzarri and Cannistraro summarize their design, establishment and possible improvement of SERS-based detection of wild-type and mutant p53. The principle, experimental methods, optimization efforts and promising results are well described in a way that non-experts can understand. This article will help attract more cancer researchers (basic, translational or clinical) to this advanced technology, likely leading to large-scale collaborative projects using cancer patients-derived serum. This reviewer has some comments and questions as follows.

Title: How about adding “toward cancer serum diagnostics” or something?

Figure 1 seems a wrong figure. There might be some confusion with Figure 11. Show the correct figure.

p53 mutations are often coincident with its overexpression, so detecting the high level itself may be of diagnostic help, even if wild-type or mutated status is unknown.

Lines 39-40 (refs 10-13): Any examples of abnormal concentrations of wild-type p53? Non-cancer diseases?

Lines 78-79 and a few other places: The word “isoforms” is not appropriate to describe mutant p53. For example, see Joruiz SM et al. The Δ133p53 Isoforms, Tuners of the p53 Pathway. Cancers (Basel). 2020 Nov 18; 12(11): E3422. doi: 10.3390/cancers12113422  

R249S is well known to be specifically induced by aflatoxin. Aflatoxin-associated hepatocellular carcinoma will be a good candidate for patients serum testing.

Line 271: Figure 5, not 6.

Figure 7: Why not dose-dependent? This data is in contrast to Fig 6 and 8.

Author Response

Title: How about adding “toward cancer serum diagnostics” or something?

The title has been slightly modified to take into consideration the Reviewer’s suggestion. The new title is: “Toward cancer diagnostics of the tumor suppressor p53  by Surface Enhanced Raman Spectroscopy”  

Figure 1 seems a wrong figure. There might be some confusion with Figure 11. Show the correct figure.

Indeed, Fig.11 and  Fig.1 were mixed. The error has been corrected (see Fig.1).

p53 mutations are often coincident with its overexpression, so detecting the high level itself may be of diagnostic help, even if wild-type or mutated status is unknown.

We agree with the Reviewer, however our approach allows detection of p53  in the 10-17M-10-12 M range, which is rather wide and it is expected to cover the main requirements for detection of p53 in clinics; moreover  it can be easily extended to reveal higher concentrations.

Lines 39-40 (refs 10-13): Any examples of abnormal concentrations of wild-type p53? Non-cancer diseases?

Indeed, wild-type p53 was found to be overexpressed in some type of cancers as it was put into evidence in ref.[14]. This aspect  has been more explicitly mentioned in the text (Pag.1 Lines 39-40).

Lines 78-79 and a few other places: The word “isoforms” is not appropriate to describe mutant p53. For example, see Joruiz SM et al. The Δ133p53 Isoforms, Tuners of the p53 Pathway. Cancers (Basel). 2020 Nov 18; 12(11): E3422. doi: 10.3390/cancers12113422  

The word isoforms was removed through the text and the mentioned reference has been added to the text (see Pag.1 Line 35 and ref.[9]).

 R249S is well known to be specifically induced by aflatoxin. Aflatoxin-associated hepatocellular carcinoma will be a good candidate for patients serum testing.

The connection between the R249S mutant and Aflatoxin-associated hepatocellular carcinoma has been now mentioned in the text (Pag.4 Lines 144-145 and the newly added reference (ref.[55])).

Line 271: Figure 5, not 6.

Done

Figure 7: Why not dose-dependent? This data is in contrast to Fig 6 and 8.

Indeed, in Figure 7 a different scale for the various spectra was used and this could lead to a misunderstanding. To avoid this, Figure 7 was re-drawn by re-scaling  the spectra (see new Figure 7).   However, we would like to remark that a quantitative relationship emerges only when the average from a collection of spectra are analysed. This aspect has been better explained in the text (Pag.8 Lines 301-304).